# Interkingdom Plant–Soil Microbial Ecological Network Analysis under Different Anthropogenic Impacts in a Tropical Rainforest

Jingjing Yu [1], Wei Cong [1], Yi Ding [1], Lixiao Jin [1], Jing Cong [2,*] and Yuguang Zhang [1,*]

1. Ecology and Nature Conservation Institute, Chinese Academy of Forestry, Key Laboratory of Biological Conservation of National Forestry and Grassland Administration, Beijing 100091, China; 15034576002@163.com (J.Y.); cong0915@163.com (W.C.); dingyi@caf.ac.cn (Y.D.); jin137762@163.com (L.J.)
2. College of Marine Science and Biological Engineering, Qingdao University of Science and Technology, Qingdao 266042, China
* Correspondence: yqdh77@163.com (J.C.); zhangyg@caf.ac.cn (Y.Z.); Tel.: +86-010-62889240 (Y.Z.)

**Abstract:** Plants and their associated soil microorganisms interact with each other and form complex relationships. The effects of slash-and-burn agriculture and logging on aboveground plants and belowground microorganisms have been extensively studied, but research on plant–microbial interkingdom ecological networks is lacking. In this study, using old growth forest as a control, we used metagenomic data (ITS and 16S rRNA gene amplified sequences) and plant data to obtain interdomain species association patterns for three different soil disturbance types (slash-and-burn, clear cutting and selective cutting) in a tropical rainforest based on interdomain ecological network (IDEN) analysis. Results showed that the soil bacterial–fungal and plant–microbe ecological networks had different topological properties among the three forest disturbance types compared to old growth forest. More nodes, links, higher modularity and negative proportion were found in the selective cutting stand, indicating higher stability with increasing antagonistic relationships and niche differentiation. However, the area of slash-and-burn forest yield opposite results. Network module analysis indicated that different keystone species were found in the four forest types, suggesting alternative stable states among them. Different plant species had more preferential associations with specific fungal taxa than bacterial taxa at the genus level and plant–microbe associations lagged behind bacterial–fungal associations. Overall, compared with old growth forests, the bacterial–fungal and plant–microbe ecological networks in the slash-and-burn and clear cutting stands were simpler, while the network in the selective cutting stand was more complex. Understanding the relationships between aboveground plants and belowground microorganisms under differing disturbance patterns in natural ecosystems will help in better understanding the surrounding ecosystem functions of ecological networks.

**Keywords:** disturbance types; community recovery; interdomain ecological network; bacterial–fungal interactions; plant–microbe association; SparCC method; high-throughput sequencing

## 1. Introduction

Human activities, such as slash-and-burn agriculture and logging, have severely damaged tropical rainforest ecosystems and biodiversity [1,2]. Some restored forests look superficially like mature rainforests, yet are still greatly lacking biodiversity, which could take centuries to fully recover [3]. In recent years, a plethora of research has been conducted on the restoration of aboveground plant diversity and underground microorganisms. Studies have found that the plant species composition [4,5] and soil microbial communities [6,7] of secondary forests were significantly different from that of old growth forest. Han et al. [8] showed that logging disturbance significantly changed the ecological strategy spectrum, and that the strategy number increased with logging intensity. Chen et al. [7] found that

the ecological network structure of the microbial community in landscapes with selective logging was more complex than those with clear cutting. Despite such work, there are few studies on interkingdom plant–microbial ecological networks during the restoration period of secondary forests.

Plants and their associated diverse soil microorganisms, such as bacteria, fungi, archaea and protists, as well as microbe–microbe pairs, interact with each other (competitive, mutualistic, neutral, commensal) and form complex relationships [9,10]. Microorganisms can protect host plants from the invasion of pathogenic microorganisms [11], boost the growth of host plants [12] and help host plants adapt to salt, drought and other adverse environments [13,14]. In exchange, the large amount of sugars, amino acids and organic acids secreted by plants in the rhizosphere are nutrient sources for soil microorganisms [15]. The structure and function of soil microbial communities is generated by the complex interactions among host plants, the surrounding plant community, the soil environment and microbial co-associations with the microbiome [9]. Interactions between these species change with the environment [16]. In this study, we took samples in non-rhizosphere soil loci. The composition of the rhizosphere microbiome is extremely different from the structure of the native soil microbiome [17], but the plant rhizosphere-specific fungal mycelia may extend into the bulk soil, affecting the non-rhizosphere microbial community [18]. Understanding the relationships between aboveground plants and belowground microorganisms under different disturbance types in natural ecosystems will aid in better understanding of the ecosystem functions of ecological networks.

With the rapid development of high-throughput technologies, microbiologists increasingly recognize that species interactions are more important than species richness in complex ecosystems [19,20]. These interactions can be visualized as a series of ecological networks in which species are directly or indirectly linked [19]. The ecological network of microbial communities has been extensively studied. For example, Deng et al. [21] constructed molecular ecological networks using Random Matrix Theory methods to provide better understanding of network interactions in microbial communities and their responses to environmental changes. Ecosystem functions of ecological networks can be identified based on network topology [20]. In order to better understand the associations between aboveground plants and belowground microorganisms, Feng et al. [22] proposed a workflow to construct interdomain ecological networks between plants and microbes. Chen et al. [23] demonstrated that interdomain ecological networks between plants and microbes can reflect changes in tree composition and soil nutrients during forest restoration.

Hainan Island hosts the largest tropical forest in China and is one of the world's biodiversity hotspots. Due to the interference of human activities, such as slash-and-burn agriculture and logging, a variety of secondary forests have been formed during the natural recovery process, making the composition of plant species significantly different from that of old growth forests [5], which provides an ideal experimental platform for our research. In this study, using old growth forest (OG) as a control, three types of disturbed forest land (slash-and-burn (SB), clear cutting (CC) and selective cutting (SL)) were used to construct ecological networks between bacteria and fungi as well as plants and microbes to address the response of soil bacterial–fungal and plant–microbe ecological networks to different disturbance types.

## 2. Materials and Methods

### 2.1. Study Area and Soil Sampling

The study area is located in a tropical lowland rain forest in the Bawangling Nature Reserve (BNR), which is in southwest Hainan Province, China (19°04′~19°08′ N, 109°07′~109°10′ E). The study area is mountainous, with elevations ranging from 100 m to 1654 m a.s.l., having a tropical monsoon climate with the dry season from November to April and wet season from May to October. The mean annual temperature (MAT) is 22.3 °C and mean annual precipitation (MAP) is 2422.7 mm (average from 2012 to 2018 (years)) [24]. There are currently four types of forest stands in BNR: the old growth forest stand (OG),

without logging history, and the secondary forest stand, which has naturally regenerated for 60 years (at most) after slash-and-burn (SB) agricultural practices and 40 years after clear cutting (CC) and selective cutting (SL) practices.

The samples were collected from October to December, 2018. Twelve 20 m × 20 m quadrats per forest stand were established at intervals of approximately 20 m. A soil drill with a diameter of 3 cm was used to collect 16 soil samples with a depth of 0–10 cm according to the diagonal principle. The specific site information was shown in Table A1. After removing fine roots, debris and litter, the soil was sieved through 2 mm mesh and brought back to the laboratory for further analysis. Part of soil sample was stored at −20 °C for soil DNA extraction, and the other part was stored at 4 °C for soil physicochemical analyses.

### 2.2. Soil Physicochemical Parameters and Plant Survey

Soil physicochemical parameters were measured as previously described [25]. Soil pH was measured at a mass ratio of 1:2.5 (soil:water) using a pH meter with a calibrated combined glass electrode. Soil organic carbon (SOC) was measured using the potassium dichromate volumetric and external heating. Total nitrogen (TN) was measured using the modified Kjeldahl procedure. Total phosphorus (TP) and total potassium (TK) were measured using automatic digestion apparatus and plasma emission spectrometer. Available phosphorus (AP) was measured at ammonium fluoride ($0.03$ mol·L$^{-1}$)-hydrochloric acid ($0.025$ mol·L$^{-1}$) extraction solution using continuous flow analyzer. Available potassium (AK) was measured at 1 mol·L$^{-1}$ ammonium acetate extraction using plasma emission spectrometer. Soil $NH_4^+$-N and $NO_3^-$-N were measured at 2 mol·L$^{-1}$ potassium chloride extraction using continuous flow analyzer. The soil physicochemical parameters were significantly by different anthropogenic impacts (Table A2).

All woody plants with a diameter at breast height (DBH) $\geq 1$ cm were recorded and identified to species. Height, DBH, coordinates and other information were recorded. Compared with the OG, the plant Shannon and richness indexes of SB were significantly decreased, while that of the CC and SL were significantly increased ($p < 0.05$) (Table A2). The important value index (IVI = (relative frequency + relative density + relative basal area)/3) of species was calculated to detect the dominant species. *Koildepas hainanense*, *Castanopsis tonkinensis*, *Vatica mangachapoi* and *Cyclobalanopsis patelliformis* are the most dominant species in OG forest. While SB is dominated by *Engelhardia roxburghina*, *Psychotria rubra* and *Liquidambar formosana*, CC is dominated by *P. rubra*, *Castanopsis carlesii* var. *spinulosa* and *Adinandra hainanensis*, and SL is dominated by *Koilodepas hainanense*, *C. tonkinensis* and *Canarium album* (Table A3).

### 2.3. Soil DNA Extraction, Illumina Sequencing and Data Processing

Genomic DNA representative of the soil's microbial community was extracted from each of the 0.25 g soil samples using the PowerSoil kit (Qiagen, Germany). Extracted DNA was checked on a 2% agarose gel, and DNA concentration and purity were determined using a NanoDrop 2000 spectrophotometer (Thermo Fisher Scientific, Waltham, MA, USA). The hypervariable segment V3-V4 of the bacterial 16S rRNA gene was amplified with forward primer pairs 338F (5′-ACTCCTACGGGAGGCAGCA-3′) and reverse primer 806R (5′-GGACTACHVGGGTWTCTAAT-3′) [26], and the ITS1 segment of the fungi ITS amplified with forward primer ITS1F (5′-CTTGGTCATTTAGAGGAAGTAA-3′) and reverse primer TIS2R (5′-GCTGCGTTCTTCATCGATGC-3′) [27] using a thermocycler (ABI, Carlsbad, CA, USA). The PCR amplification was performed in 20-μL reactions and in triplicate. PCR products were extracted from the 2% agarose gel and purified using the AxyPrep DNA Gel Extraction Kit (Axygen Biosciences, Union City, CA, USA) according to manufacturer's instructions. Purified amplicons were pooled in equimolar ratios and sequenced on an Illumina MiSeq PE300 platform (Illumina, San Diego, CA, USA) using paired-end sequencing according to the standard protocols.

Raw sequences were demultiplexed, quality-filtered using fastp version 0.20.0 [28] and merged using FLASH version 1.2.7 [29]. Operational taxonomic units (OTUs) with a 97% similarity cutoff [30,31] were clustered using UPARSE version 7.1 [31]. Chimeric sequences were identified and removed. The taxonomy of each representative OTU sequence was analyzed using RDP Classifier version 2.2 with a confidence threshold of 0.7 [32].

*2.4. Network Construction*

Bacterial–fungal networks were constructed based on the Random Matrix Theory (RMT) algorithm using the Molecular Ecological Network Analyses Pipeline (MENA) (http://ieg4.rccc.ou.edu/mena/, accessed on 5 April 2022) [20,21]. The OTU tables of bacteria and fungi for each forest type were combined into one table. A total of 50% of OTUs from all samples were kept to construct the network. The same threshold was selected to compare network topology characteristics among different forest types. Network connections were randomly re-wired to calculate random network properties and make comparisons to the empirical network.

The plant–bacteria and plant–fungi interdomain ecological networks (IDEN) were constructed based on data from Sparse Correlations for Compositional data (SparCC) using a Galaxy-based analysis pipeline (integrated Network Analysis Pipeline, iNAP, http://mem.rcees.ac.cn:8081, accessed on 5 April 2022) [22] with three main steps. (1) IDEN construction: 80% of the sample numbers were kept to filter plant species or OTUs that were less detectable among all samples. The SparCC method was used to calculate the pairwise correlation value for plants and microorganisms [33]. The threshold value was set to 0.3 to filter noncorrelated associations. The finally resulting adjacent matrix consisted of values of 1 or 0, indicating the presence or absence of a plant–microbe association. (2) Network properties and visualization. The network properties included connectance, web asymmetry, cluster coefficient, nestedness and modularity. The simulated annealing (slow) method was also used for module separation and module hubs. The topological function of each node was evaluated using among-module connectivity (Pi) and within-module connectivity (Zi) [34]. Each node can be divided into four functional groups [21], namely, peripherals ($Zi \leq 2.5$ and $Pi \leq 0.62$), connectors ($Zi \leq 2.5$ and $Pi > 0.62$), module hubs ($Zi > 2.5$ and $Pi \leq 0.62$) and network hubs ($Zi > 2.5$ and $Pi > 0.62$). Module separation was visualized using Gephi (0.9.2). (3) calculating the random network properties, and measuring significance of the observed networks using a one-sample Student's *t*-test.

## 3. Results

*3.1. Topological Characteristics for Soil Bacterial–Fungal Ecological Networks*

Soil bacterial–fungal ecological networks showed clearly different topological properties among the three forest stands when compared to old growth forest (Table 1). Compared to the OG network, CC and SL networks were more complex with differing bacterial–fungal interactions, and a more modular structure, while the SB network was simpler. For example, SB had 17% fewer nodes and 46% fewer links than OG, while CC had 2 times the number of nodes and 1 to 2 times the number of links, and SL had 15 times the number of nodes and 5 to 6 times the number of links, which comparted to OG. A higher positive proportion of bacterial–fungal interactions were found in SB (93.75%) and CC (69.43%), while a lower positive proportion was found in SL (51.14%) relative to OG (56.80%). The modularity of SB (0.390) was lower than OG (0.526), while a higher modularity was observed in CC (0.722) and SL (0.897).

**Table 1.** Soil bacterial–fungal topological characteristics in the four forest stands.

| Topological Characteristics | SB | CC | SL | OG |
|---|---|---|---|---|
| **Empirical network** | | | | |
| Threshold | 0.890 | 0.890 | 0.890 | 0.890 |
| Total nodes | 57 | 139 | 1067 | 69 |
| Total links | 112 | 229 | 1187 | 206 |
| Positive interactions proportion | 93.75% | 69.43% | 51.14% | 56.80% |
| $R^2$ of power-law | 0.406 | 0.884 | 0.961 | 0.601 |
| Average degree (avgK) | 3.930 | 3.295 | 2.225 | 5.971 |
| Average clustering coefficient | 0.027 | 0.189 | 0.053 | 0.385 |
| Average path distance | 2.923 | 6.068 | 8.424 | 2.246 |
| Average geodesic distance | 2.328 | 4.535 | 6.756 | 1.838 |
| Modularity | 0.390 | 0.722 | 0.897 | 0.526 |
| **Random network** | | | | |
| Average clustering coefficient | 0.099 ± 0.022 | 0.033 ± 0.011 | 0.002 ± 0.001 | 0.177 ± 0.022 |
| Average path distance | 2.930 ± 0.093 | 3.958 ± 0.101 | 6.622 ± 0.120 | 2.590 ± 0.057 |
| Average geodesic distance | 2.477 ± 0.061 | 3.408 ± 0.062 | 5.914 ± 0.087 | 2.251 ± 0.030 |
| Modularity (fast greedy) | 0.381 ± 0.014 | 0.536 ± 0.009 | 0.794 ± 0.005 | 0.295 ± 0.010 |

SB: slash-and-burn. CC: clear cutting. SL: selective cutting. OG: old growth forests.

### 3.2. Topological Characteristics for Plant–Microbe Ecological Networks

Through the IDEN construction process, an average of 10 plants and 557 bacterial OTUs were chosen to construct the plant–bacteria network, and an average of 10 fungal plants and 90 OTUs were chosen to construct the plant–fungi network (Table 2). The plant–bacteria and plant–fungi networks of SL showed more complexity than OG, including more plants (12) and microbes (bacteria:612; fungi:256), while SB and CC networks were simpler. Different disturbances had a greater impact on plant–fungi networks and the recovery process was slower in comparison to plant–bacteria networks.

**Table 2.** IDEN topological characteristics for plant–microbe ecological networks.

| Topological Characteristics | Plant–Bacteria Network | | | | Plant–Fungi Network | | | |
|---|---|---|---|---|---|---|---|---|
| | SB | CC | SL | OG | SB | CC | SL | OG |
| No. plants | 9 | 9 | 12 | 10 | 9 | 9 | 12 | 10 |
| No. microbes | 506 | 502 | 612 | 608 | 47 | 70 | 129 | 112 |
| Total links | 1060 | 1121 | 1051 | 1221 | 93 | 178 | 256 | 240 |
| Links per species | 2.058 | 2.194 | 1.684 | 1.976 | 1.661 | 2.253 | 1.816 | 1.967 |
| Linkage density | 69.793 | 76.680 | 54.237 | 78.477 | 7.387 | 14.831 | 15.082 | 14.042 |
| Connectance | 0.232 | 0.248 | 0.143 | 0.201 | 0.220 | 0.283 | 0.165 | 0.214 |
| Web asymmetry | −0.965 | −0.965 | −0.962 | −0.968 | −0.679 | −0.772 | −0.830 | −0.836 |
| Cluster coefficient | 0.178 | 0.205 | 0.139 | 0.174 | 0.213 | 0.257 | 0.151 | 0.210 |
| Nestedness | 44.830 | 38.614 | 38.066 | 41.888 | 44.831 | 29.048 | 35.357 | 53.212 |
| Weighted nestedness | 0.189 | 0.287 | 0.216 | 0.247 | 0.002 | 0.343 | 0.222 | 0.086 |
| Specialization asymmetry | 0.666 | 0.637 | 0.653 | 0.608 | 0.435 | 0.425 | 0.519 | 0.637 |
| Modularity | 0.411 | 0.368 | 0.490 | 0.420 | 0.473 | 0.328 | 0.475 | 0.450 |
| No. of modules | 4 | 3 | 4 | 5 | 5 | 5 | 6 | 4 |

SB: slash-and-burn. CC: clear cutting. SL: selective cutting. OG: old growth forests.

There were some basic topological characteristics in the plant–microbe networks, such as connectance, web asymmetry, specialization asymmetry and modularity. The plant–microbe networks showed a non-nested structure. The average web asymmetry of the plant–bacteria and plant–fungi networks were −0.965 and −0.779, illustrating a more skewed pattern for plant–bacteria than plant–fungi. The specialization asymmetry of plant–bacteria networks was generally higher than plant–fungi networks in the four forests, and the specialization asymmetry of plant–bacteria networks increased, while that of the plant–fungi networks decreased after disturbance, suggesting that fungi had a higher preference to specific plant species than bacteria. Comparison of the observed and random

networks indicated that the observed topological characteristics of the plant–microbe had nonrandom features in the four forests (Table A4). For example, there was a significant difference for non-nested structure and modularity (*p* < 0.001).

### 3.3. Plant–Microbe Module Separation Analysis

Module separation analysis showed plant–microbe specific modules in the four forests (Figure 1). The nodes of the plant–bacteria and plant–fungi networks were relatively evenly distributed in the four forest types. For example, in the plant–bacteria network, SB had four modules, of which module #1 contained 3 plants and 149 OTUs, module #2 contained 2 plants and 115 OTUs, module #3 contained 2 plants and 86 OTUs and module 4 contained 2 plants and 156 OTUs. In the plant–fungi network, SB had five modules, of which module #1 contained 2 plants and 8 OTUs, module #2 contained 2 plants and 8 OTUs, module #3 contained 1 plant and 10 OTUs, module #4 contained 2 plants and 12 OTUs and module #5 contained 2 plants and 9 OTUs.

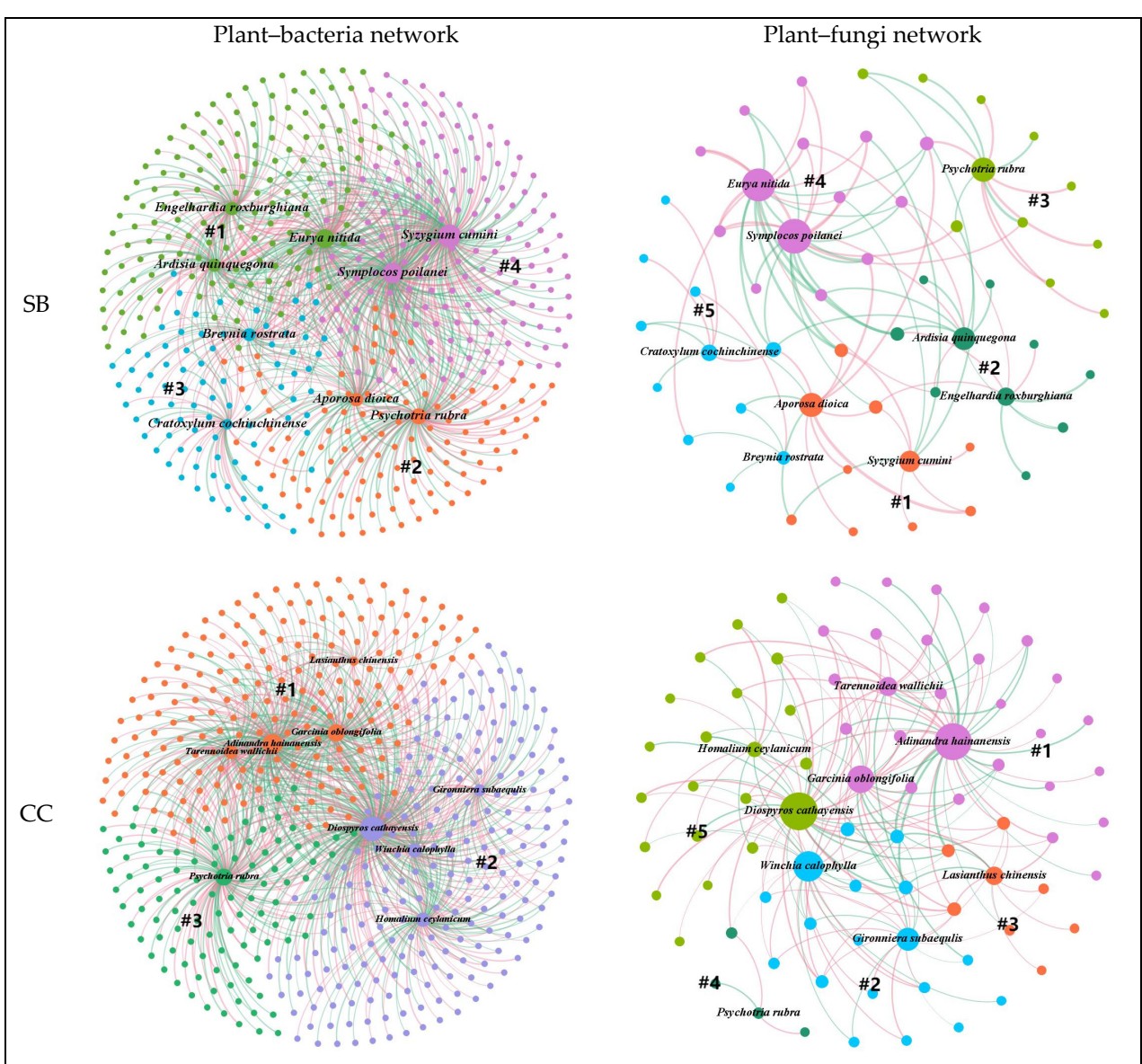

**Figure 1.** *Cont.*

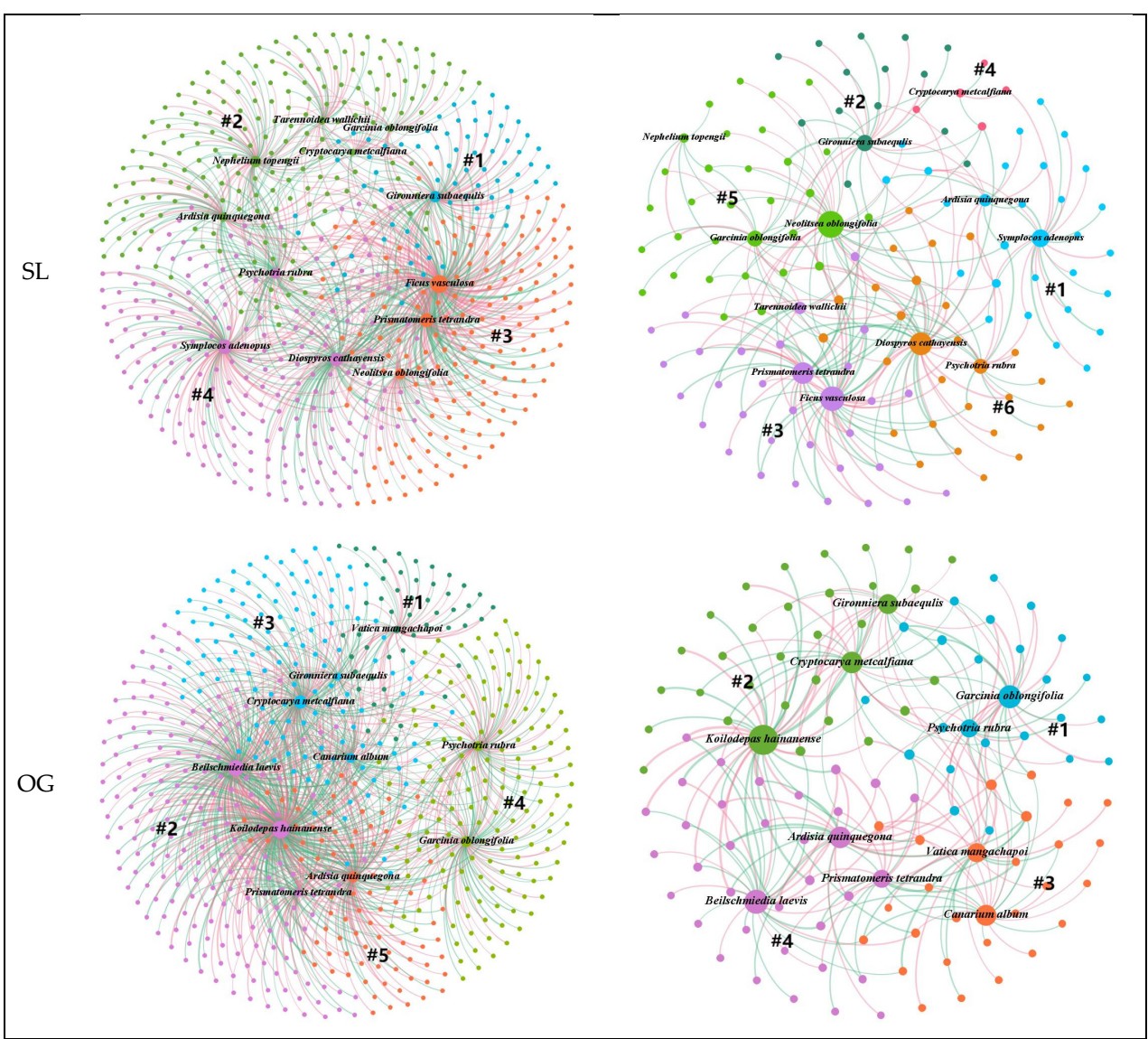

**Figure 1.** The plant–microbe module separation results. Different colors represent different modules and are represented by # numbers. The node size represents the node degree. SB: slash-and-burn. CC: clear cutting. SL: selective cutting. OG: old growth forests.

Most plants in the plant–microbe networks were dominant species (Figure 1, Table A3), suggesting that these plant species were more important. For example, *Engelhardia roxburghiana* (IVI = 0.133), *P. rubra* (IVI = 0.083), *Cratoxylum cochinchinense* (IVI = 0.029) and *Aporosa dioica* (IVI = 0.027) were the dominant species in the plant–microbe network of SB. *P. rubra* (IVI = 0.056), *A. hainanensis* (IVI = 0.030), *Diospyros cathayensis* (IVI = 0.029) and *Lasianthus chinensis* (IVI = 0.026) were the dominant species in the plant–microbe network of CC. *D. cathayensis* (IVI = 0.019), *Nephelium topengii* (IVI = 0.018), *Ficus vasculosa* (IVI = 0.017) and *Gironniera subaequlis* (IVI = 0.015) were the dominant species in the plant–microbe network of SL. *K. hainanense* (IVI = 0.090), *V. mangachapoi* (IVI = 0.036), *Ardisia quinquegona* (IVI = 0.029) and *C. album* (IVI = 0.019) were the dominant species in the plant–microbe network of OG.

Further analysis of modular roles indicated that plant species were classified as module hubs in the plant–bacteria network after different disturbances (Figure 2a–d), while most plant species were classified as peripherals in the plant–fungi network of SB and CC (Figure 2e,f), suggesting that these plant species were less important in plant–fungi

networks. More plant species were classified as module hubs in the SL plant–fungi network (Figure 2g), suggesting that these plant species were important in plant–fungi network. This proves that different disturbances can have greater impact on plant–fungi networks than plant–bacteria networks. In addition, while SB and CC showed decreased plant–microbe network complexity, SL increased in complexity.

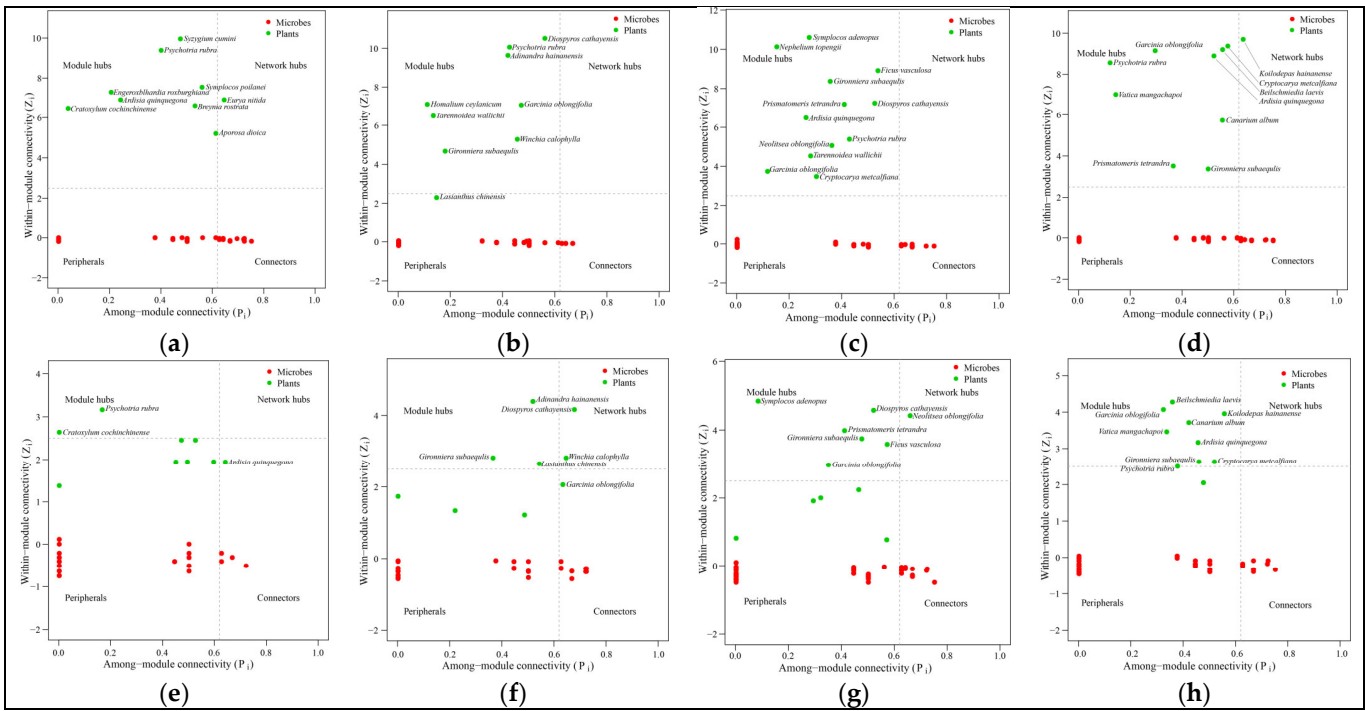

**Figure 2.** Modular roles of plant species and microorganisms based on Z-P value. (**a**–**d**) and (**e**–**h**) were Z-P plot of the plant–bacteria and plant–fungi networks of SB (slash-and-burn), CC (clear cutting), SL (selective cutting) and OG (old growth forests), respectively.

### 3.4. Plant–Microbe Genera Associations

In the four forest stands, an average of eight bacterial phyla were associated with plants, mainly consisting of two phyla (Proteobacteria and Acidobacteria) (Figure 3). There were three fungal phyla associated with plants, mainly Ascomycota (Figure 4). The plants classified as hubs (module hubs and network hubs) showed associations with more microorganisms. In the plant–bacteria network, most plants were classified as module hubs or network hubs in the four forest stands (Figure 2a–d), and each plant was associated with microorganisms (Figure 3). In the plant–fungi network, *A. hainanensis*, *D. cathayensis*, *Winchia calophylla* and *G. subaequlis* were classified as module hubs or network hubs in the CC forest stand (Figure 2f), most plants were classified as module hubs or network hubs in the SL and OG forest stands (Figure 2g,h), and these plants was associated with most microorganisms (Figure 4).

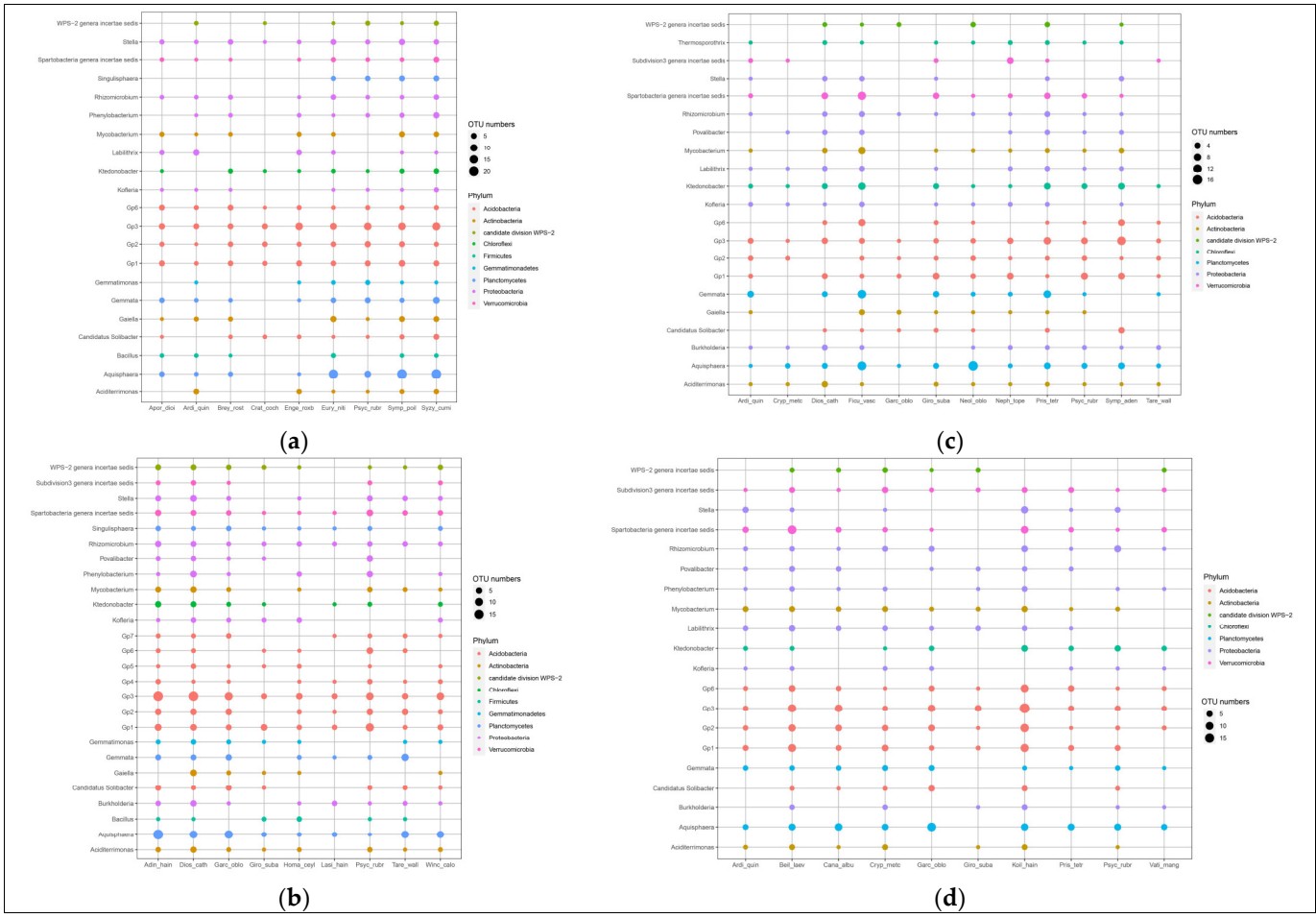

**Figure 3.** Bacteria genera associated with plants. The different colors indicate bacterial taxonomy at phylum level. (**a**–**d**) represent SB (slash-and-burn), CC (clear cutting), SL (selective cutting) and OG (old growth forests) forest stands, respectively.

Different plant species had preferential associations with specific fungal taxa over bacterial taxa at the genus level. In the plant–bacteria network, there were 13 bacteria genera that generally showed consistent associations with most plant species in the four forest types, including Proteobacteria (Stella, Rhizomicrobium and Kofleria), Acidobacteria (Gp1, Gp2, Gp3, Gp6 and Candidatus solibacter), Actinobacteria (Mycobacterium, Gaiella and Aciditerrimonas), Planctomycetes (Gemmata and Aquisphaera), Chloroflexi (Ktedonobacter) and Candidate division WPS−2 (WPS−2 genera incertae sedis). In the plant–fungi network, the genera Sphaerostilbella and Metarhizium were only associated with *A. dioica*, Aspergillus was only associated with *Breynia rostrata*, and Talaromyces, Clitopilus and Chaetosphaeria were only associated with *P. rubra* in the SB forest stand (Figure 4a). The genera Pestalotiopsis and Cladophialophora were only associated with *A. hainanensis*, Phialocephala, Metacordyceps and Archaeorhizomyces were only associated with *D. cathayensis*, and Humicola and Gliocephalotrichum were only associated with *L. chinensis* in the CC forest stand (Figure 4b). The genera Metacordyceps was only associated with *Neolitsea oblongifolia*, and Trichosporon, Ilyonectria and Cylindrocladium were only associated with *Symplocos adenopus* in the SL forest stand (Figure 4c). The genera Geminibasidium was only associated with *Beilschmiedia laevis*, and Clavaria was only associated with *Cryptocarya metcalfiana* in the OG forest stand (Figure 4d).

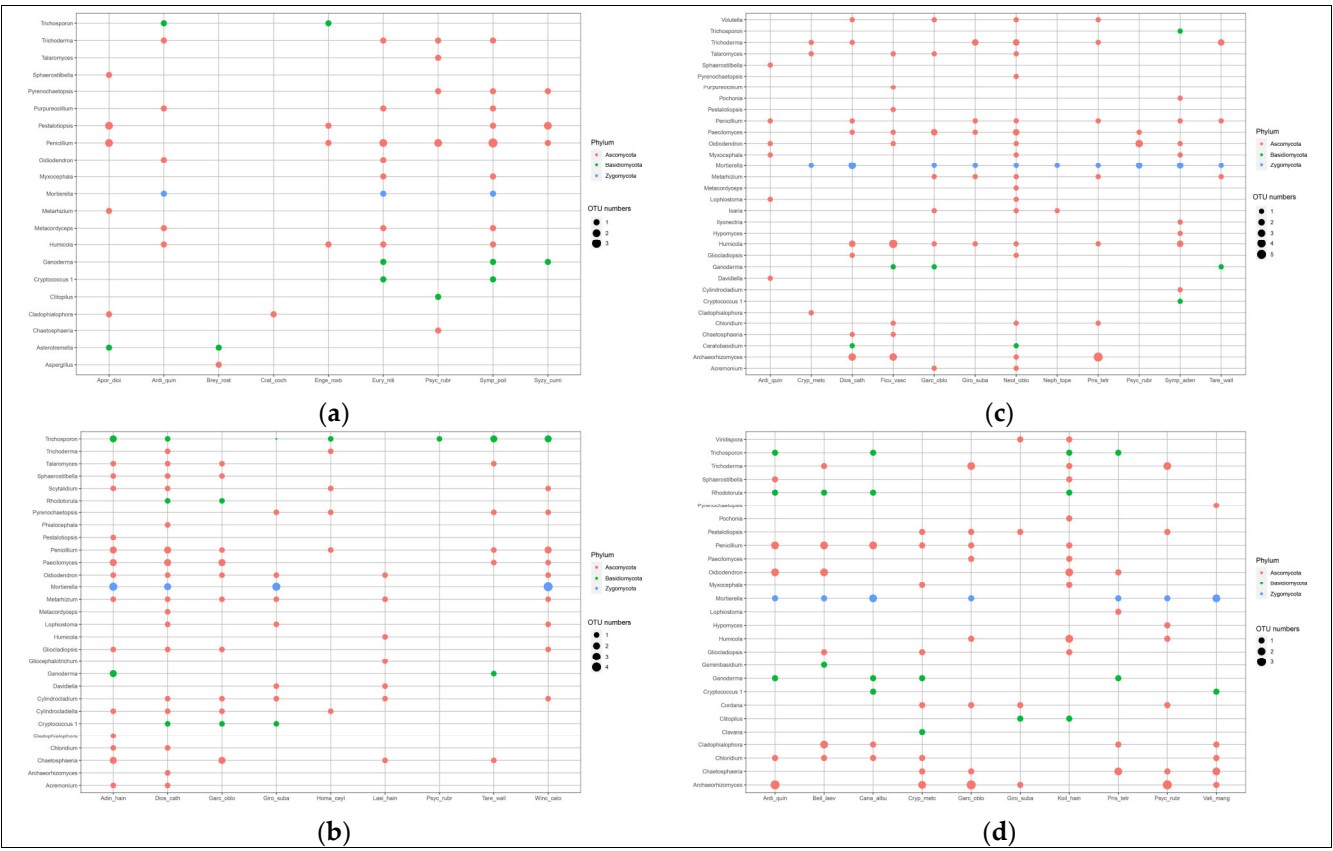

**Figure 4.** Fungi genera associated with plants. The different colors indicate fungal taxonomy at phylum level. (**a**–**d**) represent SB (slash-and-burn), CC (clear cutting), SL (selective cutting) and OG (old growth forests) forest stands, respectively.

## 4. Discussion

Molecular ecological network analysis simplifies the complex process of analyzing large amounts of data and aids in better understanding the relationships between microorganisms. The stability of a molecular network is closely related to the complexity of the network—the more complex the microbial community, the more stable the microbial community [35,36]. Diversity of interaction types (moderate mixture of antagonistic and mutualistic interactions) can stabilize population dynamics [35], indicating that soil microbial community stability is higher in the selective cutting and clear cutting stands than in old growth forest. In contrast, the stability of the soil microbial community was lower in the slash and burn stands than in old growth forest. In this study, the positive proportion of bacterial–fungal networks in slash-and-burn stands was as high as 93%, indicating an increased mutualism between bacteria and fungi. The increased proportion of negative correlation in selective cutting stands indicates an increase in antagonistic interactions between bacteria and fungi, which reflect the competition between soil bacteria and fungi for limited resources within a unique environmental niche [36]. Modularity prevailed in the molecular ecological network [21]. A network module is a group of species that are highly associated with each other, but had much fewer associations with other modules species [21], which could be perceived as niches [37]. In the bacterial–fungal networks, a higher modularity was observed in the clear cutting and selective cutting stands, suggesting niche differentiation (i.e., the microbial community segregated into finer niches and functional units). Such niche differentiation is an important factor in shaping interactive species and maintaining biodiversity at different scales [38].

Different from bacterial–fungal networks, the plant–bacteria and plant–fungi ecological networks of the clear cutting stand was relatively simple. Clear cutting drastically

changes the stand structure and removes the forest canopy [39], leading to the plant community's slow recovery, suggesting that plant–microbe associations lag behind bacterial–fungal associations. Coexistence of nested and modularity properties is crucial for the resilience and structural stability of ecosystem communities [40]. Nestedness depends on the neutral theory of community construction, while modularity depends on local adaptation and competition [41]. Studies have found that modularity is strong in trophic networks, while nested structures are strong in mutualistic networks [42]. Low nestedness is very important for resilience and structural stability of ecological communities [43]. Non-nested structures have been observed both in plant–bacteria and plant–fungi ecological networks, which may be due to the both plant–bacteria and plant–fungi ecological network belonging to trophic networks [42], and this may be a typical feature of interdomain ecological networks [22], indicating that species preferences are divided to avoid competition and thus favor system-wide resource allocation [44].

Ecological networks are usually divided into modules of closely interacting species, which are connected by some supergeneralist that serves as a hub or connector species [45]. Most microorganisms are within plant-related modules rather than across modules, suggesting a specialized association between plants and microorganisms [22]. A species defined as a hub is highly associated with other species and has a strong influence on community structures [21]. Removing the hub (keystone species/taxa) can cause drastic changes in community structure and function [46]. The loss of a keystone species leads to a loss of function, and in the case of functional redundancy, observed differences in the taxonomic composition of microbial communities do not imply a loss of function [47]. In this study, different keystone species were found in four forest stands, suggesting that there may be functional redundancy, leading to an alternative stable state [46]. The significant reduction in plant species and the change in dominant species will drastically change the relationship between plants and microorganisms [48]. Some different plant species are associated with different microbial genera in the four different forest stands, suggesting that plants adapt dynamically by regulating their microbiomes. In addition, it appears that keystone species can use differing strategies to shape their microbiomes to their advantage [46,47]. Different plant species had more preferential associations with specific fungal genera than bacterial genera, suggesting that bacteria may also have functional redundancy.

The genera of bacteria associated with plants belonged primarily to Proteobacteria and Acidobacteria, while the genera of fungi mainly belonged to Ascomycota, which are dominant phyla in forest soils [49–51], indicating that these taxa are widely involved in ecological processes. The relationship between different plants and specific fungal communities are stronger than that between different plants and bacterial communities, which may be because the soil bacterial community structure is influenced by time (succession), and the characteristics of plant species and functional groups are the driving factors affecting soil fungal community structure, which can explain the large proportion of changes in fungal community composition [52]. Urbanová et al. [53] showed that 35–37% of fungal OTUs showed a preference for specific plant species, while 80% of bacterial OTUs shared multiple plant species, which was consistent with our results, indicating that different plants have a preference for fungi over bacteria. Ectomycorrhizal fungi OTUs usually show strong host preference [53]. Chen et al. [23] found that the dominant plant species (Cyclobalanopsis, Lithocarpus and Castanopsis) characterized by ectomycorrhizal fungi increased after clear cutting and selective cutting, indicating that the accumulation of mycorrhizal fungi are beneficial to plant community restoration. Genre et al. [54] showed that mycorrhizal is one of the most important biosphere interactions, providing host plants with mineral nutrients, such as phosphorus and nitrogen, important nutrients to consider for forest management after forest fire disturbance and restoration of mining areas. However, in this study, the genera of fungi associated with plants mainly belonged to Ascomycota, which may be because most ectomycorrhizal plants associate with many unrelated ectomycorrhizal fungi, and many unrelated ectomycorrhizal fungi associate with different plant hosts [55]. For example, Morris et al. [56] found that Ascomycota were more frequent and diverse on

*Quercus douglasii*, which can increases the habitats in which the plants can live, because seedlings dispersing will not be limited in new settings, and may find compatible unrelated ectomycorrhizal fungi [57].

## 5. Conclusions

Based on the analysis of interdomain ecological network, taking the undisturbed old growth forest as a control, this study constructed plant–microbial interdomain species association patterns for different disturbance types (slash-and-burn, clear cutting and selective cutting). Compared to old growth forest, the ecological network stability of slash and burn and clear cutting decreased, while the ecological network stability of selective cutting increased and showed evidence of niche differentiation. The recovery of the plant–microbial ecological network lagged behind that of microorganisms. The associations between different plants and specific fungal groups were stronger than those between plants and bacterial groups.

**Author Contributions:** Conceptualization, J.C. and Y.Z.; methodology, J.Y., W.C., Y.D. and Y.Z.; software, J.Y. and W.C.; validation, Y.Z.; formal analysis, J.Y., W.C. and L.J.; investigation, J.Y., W.C., Y.D., L.J. and Y.Z.; data curation, J.Y.; writing—original draft preparation, J.Y.; writing—review and editing, J.Y., W.C., J.C. and Y.Z.; visualization, J.Y.; supervision, Y.Z.; project administration, Y.Z.; funding acquisition, Y.Z. All authors have read and agreed to the published version of the manuscript.

**Funding:** This research was funded by the Public Welfare Project of the National Scientific Research Institution, grant number CAFYBB2018SZ005 and the National Natural Science Foundation of China, grant number 31670614.

**Institutional Review Board Statement:** Not applicable.

**Informed Consent Statement:** Not applicable.

**Data Availability Statement:** The 16S rRNA and ITS gene sequencing data have been deposited in Sequence Read Archive (SRA) database with accession number of PRJNA832492 (https://www.ncbi.nlm.nih.gov/sra/PRJNA832492), accessed on 27 April 2022.

**Acknowledgments:** We would like to thank Liangjin Yao, Pengcheng Liu, and Wei Ren for field sampling and soil samplings collection, and Ye Deng provided the data analyses pipeline. We would also like to thank Charlotte Hacker for editing the English text of this manuscript.

**Conflicts of Interest:** The authors declare no conflict of interest.

## Appendix A

**Table A1.** Descriptions about community sample plots in the four forest stands.

| Sampling Site | Latitude-Longitude | Elevation/m | Disturbance Type | Recovery Time (years) |
|---|---|---|---|---|
| Nanchahe | 109°10′47.9″ E, 19°08′31.8″ N | 470 | Slash-and-Burn (SB) | 60 |
| Nanchahe | 109°10′48.2″ E, 19°08′19.1″ N | 439 | Slash-and-Burn (SB) | 60 |
| East main line 8 km | 109°07′06.4″ E, 19°06′47.3″ N | 497 | Slash-and-Burn (SB) | 60 |
| Wuliqiao | 109°07′11.9″ E, 19°06′58.3″ N | 502 | Slash-and-Burn (SB) | 60 |
| Yajia | 109°07′28.9″ E, 19°04′30.6″ N | 687 | Clear cutting (CC) | 40 |
| Yajia | 109°07′40.7″ E, 19°04′18″ N | 716 | Clear cutting (CC) | 40 |
| yajia | 109°07′45.5″ E, 19°04′15.6″ N | 800 | Selective cutting (SL) | 40 |
| Yajia | 109°07′07.3″ E, 19°04′22.9″ N | 751 | Selective cutting (SL) | 40 |
| Wuliqiao | 109°07′13.4″ E, 19°06′52.7″ N | 577 | Old growth forest (OG) | / |
| Wuliqiao | 109°07′06.4″ E, 19°06′47.2″ N | 594 | Old growth forest (OG) | / |
| Yajia | 109°08′20″ E, 19°06′19″ N | 680 | Old growth forest (OG) | / |
| Yajia | 109°08′10″ E, 19°06′13.4″ N | 682 | Old growth forest (OG) | / |

**Table A2.** Comparison of physicochemical and plant indexes in the four forest stands.

| Indexes | SB | CC | SL | OG |
|---|---|---|---|---|
| Physicochemical index | | | | |
| Soil pH | 4.8 ± 0.2 a | 4.8 ± 0.3 ab | 4.5 ± 0.3 b | 4.5 ± 0.1 b |
| Soil organic carbon (g·kg$^{-1}$) | 27.37 ± 5.38 ab | 30.51 ± 2.79 a | 26.37 ± 4.68 ab | 23.28 ± 4.54 b |
| Total nitrogen (g·kg$^{-1}$) | 2.0 ± 0.3 ab | 2.4 ± 0.3 a | 2.3 ± 0.4 a | 1.8 ± 0.3 b |
| Total phosphorus (g·kg$^{-1}$) | 0.2 ± 0.1 a | 0.1 ± 0.1 ab | 0.1 ± 0.0 ab | 0.07 ± 0.0 b |
| Total potassium (g·kg$^{-1}$) | 32.6 ± 12.0 b | 34.2 ± 4.9 ab | 31.4 ± 10.4 b | 42.6 ± 6.6 a |
| Available phosphorus (mg·kg$^{-1}$) | 1.97 ± 0.65 a | 1.46 ± 0.49 b | 1.38 ± 0.36 b | 0.87 ± 0.12 c |
| Available potassium (mg·kg$^{-1}$) | 113.70 ± 38.50 ab | 127.40 ± 47.50 a | 119.80 ± 36.30 a | 79.69 ± 14.23 b |
| NH$_4$$^+$-N (mg·kg$^{-1}$) | 10.76 ± 3.69 ab | 12.07 ± 10.23 ab | 14.75 ± 12.51 a | 5.54 ± 2.50 b |
| NO$_3$$^-$-N (mg·kg$^{-1}$) | 3.60 ± 2.36 b | 11.33 ± 8.05 a | 15.79 ± 9.08 a | 10.81 ± 4.39 a |
| Plant index | | | | |
| Plant Shannon index | 2.6 ± 0.3 c | 3.3 ± 0.3 a | 3.4 ± 0.3 a | 3.0 ± 0.4 b |
| Plant richness index | 30.17 ± 4.63 c | 57.83 ± 11.16 a | 55.67 ± 6.60 a | 47.00 ± 6.11 b |

SB: slash-and-burn. CC: clear cutting. SL: selective cutting. OG: old growth forests. Data presents the mean value ± standard deviation, different letters next to values indicate significant different ($p < 0.05$, one-way ANOVA with Tukey HSD) in the four forest stands.

**Table A3.** Importance value index (IVI) of the top 15 species in the four forest stands.

| Topological Characteristics | Species | Important Value Index (IVI) |
|---|---|---|
| SB | | |
| | *Engelhardia roxburghiana* | 0.133 |
| | *Psychotria rubra* | 0.083 |
| | *Liquidambar formosana* | 0.067 |
| | *Castanopsis carlesii* var. *spinulosa* | 0.062 |
| | *Schima superba* | 0.045 |
| | *Cyclobalanopsis kerrii* | 0.033 |
| | *Cratoxylum cochinchinense* | 0.029 |
| | *Aporosa dioica* | 0.027 |
| | *Adinandra hainanensis* | 0.026 |
| | *Symplocos poilanei* | 0.023 |
| | *Ardisia quinquegona* | 0.022 |
| | *Lithocarpus corneus* | 0.021 |
| | *Eurya nitida* | 0.018 |
| | *Canthium horridum* | 0.018 |
| | *Castanopsis hystrix* | 0.017 |
| CC | | |
| | *Psychotria rubra* | 0.056 |
| | *Castanopsis carlesii* var. *spinulosa* | 0.052 |
| | *Adinandra hainanensis* | 0.030 |
| | *Diospyros cathayensis* | 0.029 |
| | *Lasianthus chinensis* | 0.026 |
| | *Tarennoidea wallichii* | 0.023 |
| | *Itea macrophylla* | 0.019 |
| | *Homalium ceylanicum* | 0.0185 |
| | *Schima superba* | 0.018 |
| | *Ficus vasculosa* | 0.018 |
| | *Engelhardia roxburghiana* | 0.018 |
| | *Castanopsis hystrix* | 0.017 |
| | *Glochidion wrightii* | 0.014 |
| | *Machilus gamblei* | 0.014 |
| | *Blastus cochinchinensis* | 0.014 |

**Table A3.** *Cont.*

| Topological Characteristics | Species | Important Value Index (IVI) |
|---|---|---|
| **SL** | | |
| | *Koilodepas hainanense* | 0.064 |
| | *Castanopsis tonkinensis* | 0.039 |
| | *Canarium album* | 0.030 |
| | *Cyclobalanopsis patelliformis* | 0.024 |
| | *Maclurodendron oligophlebium* | 0.021 |
| | *Diospyros cathayensis* | 0.019 |
| | *Nephelium topengii* | 0.018 |
| | *Tarennoidea wallichii* | 0.018 |
| | *Polyalthia laui* | 0.018 |
| | *Ficus vasculosa* | 0.017 |
| | *Engelhardia roxburghiana* | 0.016 |
| | *Gironniera subaequalis* | 0.015 |
| | *Triadica cochinchinensis* | 0.015 |
| | *Xanthophyllum hainanense* | 0.015 |
| | *Castanopsis hystrix* | 0.014 |
| **OG** | | |
| | *Koilodepas hainanense* | 0.090 |
| | *Castanopsis tonkinensis* | 0.040 |
| | *Vatica mangachapoi* | 0.036 |
| | *Cyclobalanopsis patelliformis* | 0.035 |
| | *Ardisia quinquegona* | 0.029 |
| | *Alstonia rostrata* | 0.027 |
| | *Hancea hookeriana* | 0.026 |
| | *Lithocarpus fenzelianus* | 0.023 |
| | *Sindora glabra* | 0.022 |
| | *Canarium album* | 0.019 |
| | *Cryptocarya metcalfiana* | 0.019 |
| | *Garcinia oblongifolia* | 0.016 |
| | *Beilschmiedia laevis* | 0.016 |
| | *Castanopsis hystrix* | 0.015 |
| | *Machilus gamblei* | 0.014 |

SB: slash-and-burn. CC: clear cutting. SL: selective cutting. OG: old growth forests.

**Table A4.** Comparison of observed and random network topological characteristics in the four forest stands.

| | Plant–Bacteria Network | | | | Plant–Fungi Network | | | |
|---|---|---|---|---|---|---|---|---|
| | Observed Network | Rewiring Network | t | *p* | Observed Network | Rewiring Network | t | *p* |
| **SB** | | | | | | | | |
| nestedness | 44.830 | 42.979 ± 0.960 | −19.290 | <0.001 | 44.831 | 43.553 ± 2.903 | −4.402 | <0.001 |
| weighted nestedness | 0.189 | 0.259 ± 0.013 | 51.871 | <0.001 | 0.002 | 0.136 ± 0.041 | 32.836 | <0.001 |
| specialization asymmetry | 0.666 | 0.670 ± 0.001 | 52.325 | <0.001 | 0.435 | 0.447 ± 0.003 | 43.465 | <0.001 |
| C.score.HL | 0.624 | 0.584 ± 0.004 | −101.787 | <0.001 | 0.710 | 0.664 ± 0.012 | −39.209 | <0.001 |
| C.score.LL | 0.638 | 0.612 ± 0.002 | −123.511 | <0.001 | 0.670 | 0.653 ± 0.007 | −68.104 | <0.001 |
| modularity | 0.411 | 0.363 ± 0.011 | −44.098 | <0.001 | 0.473 | 0.436 ± 0.008 | −47.228 | <0.001 |
| **CC** | | | | | | | | |
| nestedness | 38.614 | 38.132 ± 0.774 | −6.221 | <0.001 | 29.048 | 28.630 ± 1.498 | −2.789 | <0.05 |
| weighted nestedness | 0.287 | 0.322 ± 0.012 | 29.981 | <0.001 | 0.343 | 0.405 ± 0.029 | 21.154 | <0.001 |
| specialization asymmetry | 0.637 | 0.639 ± 0.001 | 29.721 | <0.001 | 0.425 | 0.428 ± 0.003 | 11.638 | <0.001 |
| C.score.HL | 0.584 | 0.558 ± 0.004 | −59.482 | <0.001 | 0.558 | 0.486 ± 0.021 | −35.127 | <0.001 |
| C.score.LL | 0.576 | 0.563 ± 0.002 | −57.795 | <0.001 | 0.475 | 0.455 ± 0.007 | −27.998 | <0.001 |
| modularity | 0.368 | 0.334 ± 0.009 | −36.485 | <0.001 | 0.328 | 0.311 ± 0.006 | −26.466 | <0.001 |

**Table A4.** *Cont.*

| | Plant–Bacteria Network | | | | Plant–Fungi Network | | | |
|---|---|---|---|---|---|---|---|---|
| | Observed Network | Rewiring Network | t | p | Observed Network | Rewiring Network | t | p |
| **SL** | | | | | | | | |
| nestedness | 38.066 | 35.071 ± 0.850 | −35.253 | <0.001 | 35.357 | 31.261 ± 1.562 | −26.215 | <0.01 |
| weighted nestedness | 0.216 | 0.256 ± 0.010 | 40.028 | <0.001 | 0.222 | 0.278 ± 0.024 | 23.233 | <0.001 |
| specialization asymmetry | 0.653 | 0.656 ± 0.001 | 50.846 | <0.001 | 0.519 | 0.526 ± 0.002 | 31.260 | <0.001 |
| C.score.HL | 0.815 | 0.797 ± 0.003 | −58.507 | <0.001 | 0.744 | 0.711 ± 0.011 | −29.679 | <0.001 |
| C.score.LL | 0.773 | 0.761 ± 0.001 | −124.889 | <0.001 | 0.720 | 0.701 ± 0.003 | −54.947 | <0.001 |
| modularity | 0.490 | 0.473 ± 0.015 | −11.219 | <0.001 | 0.475 | 0.444 ± 0.005 | −62.551 | <0.001 |
| **OG** | | | | | | | | |
| nestedness | 41.888 | 37.941 ± 0.670 | −58.902 | <0.001 | 53.212 | 54.224 ± 2.005 | 5.047 | <0.001 |
| weighted nestedness | 0.247 | 0.327 ± 0.009 | 87.560 | <0.001 | 0.086 | 0.062 ± 0.027 | −8.591 | <0.001 |
| specialization asymmetry | 0.608 | 0.615 ± 0.001 | 83.837 | <0.001 | 0.637 | 0.636 ± 0.000 | −16.147 | <0.001 |
| C.score.HL | 0.686 | 0.655 ± 0.003 | −100.222 | <0.001 | 0.653 | 0.645 ± 0.003 | −22.335 | <0.001 |
| C.score.LL | 0.548 | 0.636 ± 0.002 | 501.858 | <0.001 | 0.683 | 0.677 ± 0.002 | −24.625 | <0.001 |
| modularity | 0.420 | 0.378 ± 0.010 | −42.209 | <0.001 | 0.450 | 0.409 ± 0.006 | −67.648 | <0.001 |

SB: slash-and-burn. CC: clear cutting. SL: selective cutting. OG: old growth forests.

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
