# Peer review of "Interkingdom Plant–Soil Microbial Ecological Network Analysis under Different Anthropogenic Impacts in a Tropical Rainforest"

_forests, doi:10.3390/f13081167_

Round 1
Reviewer 1 Report
The manuscript was "Interkingdom plant-microbial ecological network analysis under different soil disturbance types in a tropical rainforest" very well designed and written and I therefore congratulate the authors.
However, valuable information in the material and methods limits the possibility of evaluation in my understanding.
The authors indicated that Bawangling Nature Reserve (BNR) is located in a tropical monsoon zone with altitude variation between 100 to 1654 m. Therefore, it is unlikely that altitudes close to 1600 m will continue with a tropical climate, so it is necessary to indicate the altitude of the sampled locations. Also, the average temperature indicated probably does not represent the temperature at this altitude.
The time of collection of the samples was not indicated, being of great importance since the authors indicate a monsoon climate and with great variation of precipitation "dry season from November to 91 April and wet season from May to October".
Another indication is that latosol is the soil from the areas. But it is unlikely that only this type of soil occurs in a mountainous region. The exact indication of the soil can only be done with an on-site evaluation of each area.
Forest conditions were collected " 1- Control - old growth forest; 2 - Slash-and-burn; 3) Clear cutting; 4) Selective cutting. There is no indication of the distance between the areas. Information related to slope, position in the landscape and face (north – receive less light, south- receive more light, east or west), altitude and occurrence of organic horizon were not presented and would help in the study. Still, no information was provided about the soil source material (granite, basalt ???) and about the soil, such as granulometry (clay, silt and sand), chemical properties (pH, organic matter content, availability of Ca, Mg, K and P). At least pH, total organic matter and clay percentage are required. The presence of this data can confirm the influence of previous management and support the discussion. For example, slash-and-burn can result in large losses of biomass, N and S, but raises pH and bases. Such information should also be included in the introduction when indicating the slash-and-burn issue. Also, agricultural use in mountainous region can lead to erosion losses as well as changes in fertility from lime and fertilizer use. All this can somehow show up in the chemical analysis of the soil.
My knowledge in microbiology is very limited and I have no basis for evaluating this part. But the results and discussion seem logical and well written.
Reviewer 2 Report
The presented article is devoted to the actual topic of plant-microbial interactions in anthropogenically disturbed tropical forest ecosystems. The results of the authors quite fully and in detail explain why the sustainability of ecological networks in old-growth forests is much higher than in burned and cut down forests. In addition, the study showed the reasons for the formation of stronger associations between plants and fungi, compared with associations between plants and bacteria. The studies were performed using a number of modern molecular biological methods and excellent statistical processing, so the results obtained are beyond doubt. At the same time, the article has some shortcomings: 1 - The title of the article does not fully reflect the essence of the study. For example, the authors did not consider disturbances (changes) in the soil, but analyzed only anthropogenic impacts on forest plants. Therefore, the authors should either add more information on soil transformation, or correct the title of the article. 2 - The number of keywords should be increased because they do not yet contain information about research methods. 3 - The authors did not take into account the profile structure of the soil along the horizons. In further studies, the authors should take samples not by depth, but by soil horizons, since they usually have different properties. 4 - The authors do not indicate whether they took samples in rhizosphere or non-rhizosphere soil loci. However, it is known that the composition of the rhizosphere microbiome is extremely different from the structure of the native soil microbiome. Therefore, the authors should read and cite the article by Semenov et al.: Semenov, M. V., Nikitin, D. A., Stepanov, A. L., & Semenov, V. M. (2019). The structure of bacterial and fungal communities in the rhizosphere and root-free loci of gray forest soil. Eurasian Soil Science, 52(3), 319-332. https://doi.org/10.1134/S1064229319010137 5 - It is not clear why even the old-growth forests studied by the authors of the work are predominantly associated only with Ascomycetes, but not with Basidiomycetes? Usually, a mature community is characterized by the predominance of Basidiomycete fungi. Perhaps the authors should expand the discussion on this issue. 6 - The authors give reasoning about the ecological and indicator functions of the soil and the role of microorganisms in it. I believe, therefore, the authors should read and cite the work of Nikitin and co-authors: Nikitin, D. A., Semenov, M. V., Chernov, T. I., Ksenofontova, N. A., Zhelezova, A. D., Ivanova, E. A., Khitrov, N. B. & Stepanov, A. L. (2022). Microbiological Indicators of Soil Ecological Functions: A Review. Eurasian Soil Science, 55(2), 221-234. https://doi.org/10.1134/S1064229322020090
Round 2
Reviewer 1 Report
It was included in the manuscript many of information required.
The phrase “The mean annual temperature (MAT) is 22.3℃ and mean annual precipitation (MAP) is 2422.7 mm (average from 2012 to 2018)[24]” can be confusing. It is necessary to indicate that 2012 to 2018 means period in years.
It would be very important to specify the methodology followed for determination. For example: pH can be determined in water in relation to 1:2.5 (soil:water) or 1:1 (soil:water). The pH can also be determined with 0.01 M CaCl2 in ratio (1:2.5), being generally lower than water. If the pH values ​​presented were obtained in water, they will be very low, favoring the predominance of fungi in the soil. At the same time, it favors the predominance of ammonium over nitrate.
The available P value can also be obtained by different extraction methodologies with Olsen, Bray, Mehlich 1 or 3, resin and others. It can also be determined by different types of equipment. The same for organic carbon which can be determined by different methods. That is, it would be important to specify the extraction and determination methodologies for each parameter or indicate the manual used for analytical determination.
I believe that a fundamental parameter is missing that the soil texture (clay, silt and sand contents).
It is necessary to confirm the values ​​of total K that are very high, in contrast to low values ​​of total P.
Use the same number of decimal places to represent total P
Table A1. Descriptions about community sample plots in the four forest stands.
Recovery time (years).
